# Feasibility of Elder-Friendly Food Applications of Sacha Inchi According to Cooking Method: Focusing on Analysis of Antioxidative Activity and Brain Neuron Cell Viability

**DOI:** 10.3390/foods10122948

**Published:** 2021-12-01

**Authors:** Dah-Sol Kim, Nami Joo

**Affiliations:** Department of Food and Nutrition, Sookmyung Women’s University, Seoul 04310, Korea; dskim115@naver.com

**Keywords:** Sacha inchi, mousse type of elderly food, antioxidant, HT22 hippocampal brain neuron cell, cognitive ability

## Abstract

One of the objectives of this study was to determine the effect of the cooking method on the antioxidant activity of Sacha inchi (*Plukenetia volubilis*). This work was focused on the importance of performing proper cooking for studying Sacha inchi. The result of this study sheds light on preventing nutritional loss with appropriate cooking methods. Three types of cooking processes were selected: uncooked, roasted at 160 °C for 6 min, boiled at 100 °C for 13 min. The results of the present study indicated that roasted Sacha inchi is distinguishable for its high content of antioxidants (total polyphenol content 485.50 μM, total flavonoid content 0.02 μg/mL, DPPH free radical scavenging activity 33.05%, ferric reducing ability 0.19 μM). The results of the present study also indicated that Sacha inchi uniquely promotes HT22 cell viability. With roasted Sacha inchi treatment, HT22 hippocampal neuronal cell showed a significantly increased number of growth (*p* < 0.001). Results also suggest that the development of tenderized Sacha inchi could help the elderly consumers achieve their target antioxidant provision in smaller portion sizes, thus curtailing the peril of sarcopenia. The mousse type of elderly food may also change the taste of many other nut consumers as they may opt to start selling and consuming Sacha inchi. It could be in the Sacha inchi industry’s best interest to make certain all of the population’s textural favors are catered.

## 1. Introduction

With the foreseen increase in the population of older people, significant industrial and financial problems are expected, in addition to the enormous pressure on health care accommodations and related services. Furthermore, they connect with the merchandises currently available on the marketplace: the item tags neither approach their desires nor fit the consciousness about their health [1]. Especially in the elderly, as they get old, their physical function decreases, and the need for few nutrients increases. It has become fundamental for the nut market to consider the older people more and progress assured items to carry out the essentials. Increasing accessibility to safe and nutritious food is one of the visions to improve the quality of life and well-being of elderly buyers [2].

The consumption of nuts is recommended for older people as an effective way to deliver essential fatty acids (FAs) to prevent cardiovascular disease, anemia, and cancer [3]. However, eating nuts is not easy for the elderly with relatively low chewing ability. Therefore, research to reduce the nutritional imbalance of the elderly by changing the way of eating nuts is needed.

Among nuts, Sacha inchi (*Plukenetia volubilis*), with its many nutrients, can be considered as one of the best choices for consumers seeking health benefits. “Sacha” in Peru means “tree, forest, wild” in English, and “Inchi” in Peru means “peanut” [4]. This plant can be an excellent nutritional cause for seniors since its great condition bio-available antioxidants, such as carotenes, phytosterols (stigmasterol and campesterol), and tocopherols. These antioxidants are necessary materials that have the capability to save an organism from the hurt resulting from oxidative strain. Thus, there is particular profit in the existence of native antioxidants in health-giving plants, as they may support a body to maintain usual stability of reactive oxygen species. Alzheimer’s has been linked with the increase in active oxygen groups or the incapacity of the body to lessen these active species that were usually generated by the organism tissues, a procedure noted as oxidative strain. Furthermore, the various proportions of the vegetation have diverse pracademic constituents, every single with great nutritional benefits. The kernels include 35~60% lipids, 25~30% proteins, vitamin E, and poly-phenols, causing them to suitable for nutritional use. The fronds are the origin of terpenoids, saponins, and phenolic compounds (flavonoids). Phenolic compounds are representative antioxidant elements regularly observed in native elements. In the medicinal and pharmacological field, antioxidant activeness, along with antimicrobial activity, is one of the most searched for bioactivities.

However, Sacha inchi has a hard texture like other nuts, making it difficult for the elderly to consume. In addition, because there are few studies on Sacha inchi, it is difficult to know how to use it properly, which makes it less accessible to consumers.

In particular, the soft texture of food is a vital factor in increasing its intake of the elderly, and it is used as an attribute of nut quality evaluation for them. According to a recent study, it is necessary to improve elder customer accessibility by diversifying the use of Sacha inchi as their interest in enhanced and processed food products with tenderness increases [5]. Traditionally only providing subjectively described information, the ministry of food and rural affairs in Korea has now moved to quantified information about the chewable hardness of food for the elderly in three levels. The first level of hardness of the Korean Industrial Standards (KS), which is defined as a level that can be chewed by the teeth, is from 55,000 to 500,000 N/m^2^. The second level, which is stated as a level that can be chewed by gum, is 22,000~50,000 N/m^2^. The third level is a level that can be chewed with the tongue, with a hardness of less than 2000 N/m^2^. Therefore, tenderization is essential for the utilization of Sacha inchi for the elderly [6]. This study was, therefore, undertaken to develop mousse type of food products using Sacha inchi for elderly with masticatory impairments.

Since raw Sacha inchi is exceptionally bitter and astringent due to the tannin, it is usually cooked by boiling, steaming, grilling, or consumed as an oil. This is also safer because Sacha inchi contains appreciable amounts of alkaloids, saponins, and lectins, which may be toxic if consumed before cooking. Based on other food studies, various cooking methods can reduce the nutritional properties of bioactive compounds and nutraceuticals. According to some studies, the quality of plants deteriorates after they are cooked. This can be affected by cooking conditions, so its quality can be improved depending on the preparation and cooking method [7]. Although there have been studies on raw Sacha inchi seeds, there have been no studies on the effect of processing conditions on their nutritional properties, edibility, and tastefulness.

Hence, the experiment was conducted to determine the effect of the cooking method on the antioxidant effect of Sacha inchi. Then, those results were utilized to develop mousse type of food products using Sacha inchi for the elderly with masticatory impairments.

## 2. Materials and Methods

### 2.1. Sample Preparation

Sacha inchi (*P. volubilis*) seeds cultivated in Da Lat (Vietnam) in 2019 were purchased by Nathan Trading Co., Ltd. (Ubon Ratchathani, Thailand) and sorted by screening for uniform shape and color to use for research as a homogenized sample. In the process of homogenization, rotten seeds were sorted out. Selected Sacha inchi seeds (≈300 g) were kept in vacuum packaging under refrigeration at 4 °C.

### 2.2. Sample Processing Method

The Sacha inchi (≈300 g) from which the seeds were removed was homogenized and divided into three portions. The raw ones were used for the analysis in the first portion. The second was roasted in a heated pan at 160 °C for 6 min. The last portion was boiled at 100 °C for 13 min with water (1 L) to ensure softness to make the seed edible. The standard of softness was determined by pressing the boiled seeds with fingers. These optimal conditions to make Sacha inchi edible were decided by a number of preliminary experiments of temperature and time.

### 2.3. Analysis of Total Polyphenol Content

Polyphenolic compounds (1 mM) were prepared in ethanol; 40 μL of Sacha inchi extract was transferred into a 10 mL screw-cap tube. Then, 800 μL of a 10-fold diluted folin ciocalteau reagent was added into the tube and mixed well. The tube was allowed to stand for 5 min. Then, 800 μL of 7% sodium carbonate aqueous solution (*w*/*v*) was added to the tube and mixed well. The volume in the tube was made up with nano pure water (360 μL), mixed well, and then allowed to stand for 2 h at room temperature. Absorbance was read at 760 nm against the blank using a UV visible spectrophotometer (T60UV, PG instruments Ltd., Lutterworth, UK).

### 2.4. Analysis of Total Flavonoid Content

Total flavonoid contents in Sacha inchi were determined using the aluminum chloride colorimetric method described by Woisky and Salatino (1998) [8]. The appropriate dilution of extractions (0.5 mL) was briefly mixed with 1.5 mL of 95% ethanol, 0.1 mL of 10% aluminum chloride hexahydrate, 0.1 mL of 1 M potassium acetate, and 2.8 mL of deionized water. After incubation at room temperature for 40 min, the absorbance of the reaction mixture was measured at 415 nm against a deionized water blank on a UV visible spectrophotometer (T60UV, PG instruments Ltd., Lutterworth, UK).

### 2.5. Analysis of DPPH Free Radical Scavenging Activity

The antioxidant activity of Sacha inchi was measured in terms of hydrogen donating of radical scavenging ability, using the stable radical, 2,2′-diphenyl-1-picrylhydrazyl (DPPH) [9]. An ethanolic stock solution (50 μL) of the antioxidant was placed in a cuvette, and 2 mL of 6 × 10^−5^ M ethanolic solution of DPPH was added. Absorbance measurements commenced immediately. The decrease in absorbance at 515 nm was determined by spectrophotometer after 1 h for all samples. Ethanol was used to zero the spectrophotometer. The absorbance of the DPPH radical without antioxidant, i.e., the control, was measured daily. Special care was taken to minimize the loss of free radical activity of the DPPH radical stock solution [10].

### 2.6. Analysis of Ferric Reducing Ability

An automated test measuring the ferric reducing ability of plasma (FRAP), the FRAP assay, is used for assessing antioxidant power [11]. Working the FRAP reagent was prepared by mixing 25 mL acetate buffer, 2.5 mL 2,4,6-Tris(2-pyridyl)-s-triazine solution, and 2.5 mL ferric chloride solution. Then, 300 μL freshly prepared FRAP reagent was warmed at 37 °C, and 10 μL of Sacha inchi extract was added, along with 30 μL distilled water. Absorbance readings were taken after 4 min at 593 nm.

### 2.7. Analysis of Viability of HT22 Cell in the Hippocampus for Validation of Cognitive Enhancement by Roasted Sacha Inchi

#### 2.7.1. Preparation of Test Material

Roasted Sacha inchi was mixed with 40% and 70% ethanol to weight concentrations 1~100 μg/mL (1, 5, 10, 20, 40, 80, and 100 μg/mL), and extracted under conditions of 24 h, 37 °C, and 50 rpm. The lower (40%) and higher (70%) ethanol density was used as a test variable to find the adequate treatment method. In order to close off from potential contamination by impurities in advance, the test material was filtered using a 0.45 μm syringe after extraction. After that, the test material was concentrated and frozen to form a solid powder and then quantitatively dispensed. However, the cell was used by dissolving the test material in the cell culture medium for maintaining the proper form during cell treatment (approval number of an institutional review board of sookmyung women’s university in Korea: SMWU-2006-HR-061).

#### 2.7.2. HT22 Cell Cultures and Treatment

The HT22 cell line is a mouse hippocampal neuronal cell line. HT22 cells were a gift from the American-type culture collection. They were maintained in Dulbecco’s modified Eagle’s medium (DMEM), supplemented with 10% heat-inactivated fetal bovine serum, 100 units/mL penicillin, and 100 μg/mL streptomycin in the same incubator at 37 °C in a humidified atmosphere containing 5% of carbon dioxide. For the culture period, the temperature and humidity of the culture room were checked every 8 h, and the medium was changed twice a week to maintain sufficient growth of cells until the end of the test.

HT22 cells were seeded in a 75 cm^2^ flask at a density of 1 × 10^7^ cells/flask, and 24 h later (the number of cells multiplied by more than 60%), cell transfer and preparation of single-cell suspensions were performed by mild enzymatic dissociation using 3 mL of 0.25% (*w*/*v*) trypsin 0.53 mM ethylene diamine tetra-acetic acid solution. Moreover, the cells were separated by centrifugation at 125× *g* for 10 min.

HT22 cells were seeded in a 48 well plate, diluting the test material with the culture medium in concentrations 1, 5, 10, 20, 40, 80, and 100 μg/mL. Cell culture viabilities were then analyzed after 24 and 48 h. The lower and higher incubation time was a test variable to find the adequate treatment method.

#### 2.7.3. Measurement of HT22 Cell Viability

HT22 cells were first cultured and then incubated for 24 and 48 h after material treatment of the test material. The method of 3-(4,5-dimethylthiazolyl-2)-2,5-diphenyltetrazolium bromide (MTT) assay is as follows. First, wash it carefully using serum-free DMEM after removing all the cell culture medium treated with the test material. Second, remove serum-free DMEM after washing, and then apply an equal amount (200 μL) to each well of serum-free media +10% of thiazolyl blue tetrazolium blue (5 mg/mL). Third, incubate at 37 °C with 5% of carbon dioxide in a humidified chamber for 2 h. Forth, treat dimethyl sulfoxide after carefully removing the supernatant. Fifth, measure the absorbance using the 550 nm wavelength of the micro-plate reader.

### 2.8. Development of Mousse Type Braised Sacha Inchi for the Elderly

#### 2.8.1. Experimental Design

To develop the taste of the mousse type braised Sacha inchi, response surface methodology (RSM) was applied using Design Expert software (Version 11, Stat-Ease Inc., Minneapolis, MN, USA). RSM is a technique to determine design factor settings to get an optimal response. A complete factorial design was used to investigate the effects of two independent variables (the content of soy sauce and corn starch syrup) at three levels (7.5, 45.0, and 82.5 g; X1 and 8.5, 25.5, and 42.5 g; X2, respectively) (Table 1). The dependent variable was the preference of taste of braised Sacha inchi (sweetness, saltiness, nuttiness, and overall taste).

#### 2.8.2. Optimization of Preparation Method for Mousse Type Braised Sacha Inchi

Sacha inchi (≈100 g) was cleaned from crushed seeds. Then, Sacha inchi was roasted in a heated pan at 160 °C for 6 min; 200.0 g of water, 39.0 g of soy sauce, and 23.29 g of corn starch syrup was added and boiled down to absorb all liquid, but not dry, and then, froze braised Sacha inchi with 200.0 g of water at −80 °C for 24 h, ground frozen Sacha inchi in a pacojet (PJ2, HRS, Seoul, Korea).

Ground frozen Sacha inchi portion (≈37.0 g) was heated in a pot at 60 °C for 1 min; gelatin powder from 3% to 41% of total weight was added in steps of 2% and heated for 1 min to melt all gelatin powder. Subsequently, the mixture was divided among the three; each of them was poured into a mold and placed in a refrigerator to cool for 12 h.

#### 2.8.3. Analysis of Hardness of Mousse Type Braised Sacha Inchi

To determine the texture of the sample, texture profile analysis (TPA) was performed with a texture analyzer (TA-XT Express 20096, Stable microsystems Ltd., London, UK). The sample was subjected to a two-cycle compression test using a 25 kg load cell. Additionally, a 50 mm diameter cylindrical probe (pre-test speed 2 mm/s, trigger force 5 g, test speed 1 mm/s, return speed 1 mm/s, test distance 7.5 mm, time 5 s) was used to compress the samples. TPA recorded the following attributes: hardness—the maximum force required to compress the sample is indicated by the first bite, and resilience—it represents the ratio between the negative and positive forces input in the first compression.

#### 2.8.4. Sensory Evaluation of Mousse Type Braised Sacha Inchi

At first, the hedonic scale was used to determine the degree of preference scores for braised Sacha inchi (approval number of an institutional review board of Sookmyung Women’s University in Korea: SMWU-2006-HR-061). For this study, 25 trained tasting panelists were recruited and informed that they would be evaluating braised Sacha inchi. Ten samples were presented in random order, and panelists were asked to evaluate the acceptability of sweetness, saltiness, nuttiness, overall taste. Panelists were asked to evaluate preference levels using a seven-point hedonic scale (1 = dislike immensely, 4 = neither dislike nor like, 7 = like remarkably). Panelists received a tray containing the samples at room temperature (randomly coded using a three-digit number), a glass of water, and a sensory evaluation sheet. Panelists were instructed on how to evaluate the samples and were not required to expectorate or consume the entire volume served. There was an inter stimulus interval of 30 s imposed between samples. Enough space was given to handle the samples and questionnaires, and evaluation time was not constrained. No specific compensation was given to the panelists.

After findings from the first sensory evaluation (optimized recipe of braised Sacha inchi), the second sensory evaluation was conducted with 110 consumers, among them between 65 and 70 years of age, where the evaluated parameters were sweetness, saltiness, nuttiness, overall taste, through a seven-point hedonic scale for optimized braised Sacha inchi (approval number of an institutional review board of Sookmyung women’s university in Korea: SMWU-2006-HR-061). The tasting protocol was the same as with the first tasting panel.

### 2.9. Statistical Analysis

The results of this experiment were tested with a parametric analysis of variance. The test is performed using Tukey’s multiple comparison test to analyze the significant differences between the test groups if the results are significant. Statistical analysis was performed using IBM SPSS statistics (Version 23.0, GraphPad Software Inc., San Diego, CA, USA), and it is determined to be statistically significant if the *p*-value is less than 0.05 (*p* < 0.05).

## 3. Results and Discussion

### 3.1. Comparisons of Antioxidant Effects of Sacha Inchi According to Cooking Method

#### 3.1.1. Total Polyphenol Content

The results of this study indicated that the high content of total polyphenol in Sacha inchi is distinguished (Table 2). The uncooked Sacha inchi contained total polyphenol content (range = 645.31~809.92 μM, mean = 737.52 μM) significantly (*p* < 0.001) higher than when it boiled (range = 144.54~167.62 μM, mean = 159.68 μM) due to a possible flow of this element into the water. Similarly, uncooked Sacha inchi substantially contained total polyphenol content (*p* < 0.001) than when it roasted (range = 469.15~497.62 μM, mean = 485.50 μM) due to the degradation by thermal processing during roasting treatment. More notably, the total polyphenol content studied in the roasted Sacha inchi was significantly (*p* < 0.001) higher than the boiled Sacha inchi (Figure 1). It means roasting is an appropriate process for keeping total polyphenol content in Sacha inchi than boiling. Given that raw Sacha inchi cannot intake because of its astringency, the results of this study confirm that roasted Sacha inchi could also be a natural source of polyphenol.

Mbah et al. (2012) [12], working with Moringa oleifera seeds, having antioxidants, observed an effect of cooking methods (boiling and roasting) on nutrients and antinutrients content. Food processing transforms raw foods into edible forms. It also increases shelf-life, digestibility, flavor, and nutritive value, among other benefits. Various foods presuppose different processing techniques depending on the needs and end products required. Cooking has been known to bring about high complex reactions without having a direct effect on the nutritional value of food. Boiling is not only the most common cooking method but also the simplest. During boiling, the action of the heated water makes the food to get cooked, and then, it is generally thrown away. Roasting is a cooking method that uses dry heat, whether an open flame, oven, or other heat sources. Roasting usually causes caramelization or Maillard browning of a surface of the food, considered flavor enhancement. Roasting uses more indirect, diffused heat (as in the oven) and is suitable for slow cooking of food in a larger, whole piece.

#### 3.1.2. Total Flavonoid Content

In the results of this study, the substantial content of total flavonoids in Sacha inchi is distinguishable. The antioxidant, anti-inflammatory, and antiproliferative activities could be relevant to flavonoids in some studies. Moreover, antiproliferative activities in Sacha Inchi were related to terpenoids [6]. Machana et al. (2011) [13], working with extracts from Sacha inchi that had flavonoids, the result showed a reduction in the proliferation rate of HepG2 tumor cells by the early and late stages of apoptosis.

Further analysis of the roasted Sacha inchi shows significantly (*p* < 0.001) higher total flavonoid content (range = 0.020~0.023 μg/mL, mean = 0.021 μg/mL) than the raw (range = 0.004~0.005 μg/mL, mean = 0.004 μg/mL) and boiled (range = 0.011~0.012 μg/mL, mean = 0.012 μg/mL) one. This implies that roasting is an appropriate process for enhancing total flavonoid content in Sacha inchi.

Other studies have shown that roasted coffee contains naturally present antioxidants and others that are formed during the roasting process [14]. Roasted coffee may play a role in the inhibition of lipid peroxidation, free radical scavenging, metal chelation, and anti-inflammatory activity. That also may reduce the risk for the development and progression of atherosclerosis and insulin resistance, and they may decrease blood pressure.

#### 3.1.3. DPPH Free Radical Scavenging Activity

The results of this study indicated that Sacha inchi is distinguishable for its high DPPH free radical scavenging activity. A close look at the roasted Sacha inchi presents significantly (*p* < 0.001) higher DPPH free radical scavenging activity (range = 30.12~35.09%, mean = 33.05%) than the raw Sacha inchi (range = 28.92~31.29%, mean = 29.89%). In contrast, the boiled Sacha inchi had significantly (*p* < 0.001) lower DPPH free radical scavenging activity (range = 26.21~28.36%, mean = 27.12%) than the raw and roasted Sacha inchi. Sterbova et al. (2017) [15] explained the increase in antioxidant activity during thermal processing by the improvement of antioxidant properties of naturally occurring compounds or by the new compounds with antioxidant properties formed during the thermal process by Maillard reaction. Such a phenomenon has been reported by Natella et al. (2002) [16] during the roasting process of coffee beans. Similarly, high-temperature treated cashew showed higher Trolox equivalent antioxidant capacity compared to the raw one. Roasting at 160 °C for 6 min of Sacha inchi can be suitable for preserving DPPH free radical scavenging activity, and it makes the roasted Sacha inchi an alternative plant-based antioxidant in the diet. The level of a natural antioxidant in Sacha inchi has beneficial effects on health.

In conclusion, the roasting treatment of Sacha inchi caused an increase of the oxidation indicators, such as DPPH free radical scavenging activity, which seemed to provide some protection to the antioxidant activity against oxidation during its roasting treatment at high temperatures. In the roasting treatment studies, antioxidant activity from roasted Sacha inchi exhibited higher DPPH free radical scavenging activity compared to antioxidant activity from the raw and boiled ones. The treatment of roasting had an important effect on enhancing the protection against oxidation. These indicate that the potential in the roasted Sacha inchi has to serve as an alternative antioxidant source in the diet as an edible seed for cooking or through nutraceutical supplements.

Vignoli et al. (2014) observed that the roasting process causes changes in the chemical composition and biological activity of the coffee: while natural phenolic compounds may be lost, other antioxidant compounds are formed, such as Maillard reaction products [17]. Literature usually describes a decrease in the antioxidant activity of the coffee beans as the roasting degree increases, which is mainly associated with the degradation of chlorogenic acids. However, the roasting process produced melanoidins, reported as the compound responsible for the antioxidant activity in the high molecular weight fractions isolated from roasted coffee. In addition, some volatile heterocyclic compounds produced during roasting have also been described as potential antioxidants. Hydroxymethylfurfural, an intermediate compound in Maillard reaction, presented antioxidant activity similar to 2,6-dibutyl-4-methylphenol and α-tocopherol.

#### 3.1.4. Ferric Reducing Ability

As a result of this study, the Ferric reducing ability of Sacha inchi is highly distinguishable. A thorough analysis of the roasted Sacha inchi shows significantly (*p* < 0.001) higher high ferric reducing ability (range = 0.18~0.20 μM, mean = 0.19 μM) than the raw Sacha inchi (range = 0.129~0.134 μM, mean = 0.133 μM). In contrast, the boiled Sacha inchi had significantly (*p* < 0.001) lower ferric reducing ability (range = 0.12~0.13 μM, mean = 0.12 μM) than the raw and roasted Sacha inchi. This indicates that roasting is a proper process for enhancing ferric reducing ability in Sacha inchi.

In other studies, some heat-labile antinutrients, which can block the absorption of nutrients into the body, will be reduced to a great extent by heating in the application [18]. So Sacha inchi must be heat treated before consumption to prevent deleterious effects (physiological disorders such as an increase in the relative weight of the pancreas and liver, and diarrhea). As thermal processes, roasting Sacha inchi can improve tenderization of the cotyledons by increasing palatability and nutritional value or inactivating endogenous toxic factors.

Similarly, roasted maize showed significantly higher ferric, reducing antioxidant power compared to the raw one [19]. Over the years, plant foods have been rich in antioxidants phytochemicals such as phenolic compounds, ascorbic acid, carotenoids, anthocyanins, phytosterols, and policosanols, known to significantly affect human health by combating or preventing the negative effect of free radicals. Increased consumption of plant foods is related to a reduced risk of chronic diseases. Part of the antioxidant activities of plant foods is related to ferric reducing antioxidant power.

### 3.2. Comparisons of HT22 Cell Viability in the Hippocampus for Validation of Cognitive Enhancement by Roasted Sacha Inchi

#### 3.2.1. HT22 Cell Viability According to Roasted Sacha Inchi Treatment Method

Cholinergic cell loss is a major feature of multiple diseases of cognition: the severity of cognitive impairment in Alzheimer’s disease and Parkinson’s dementia is correlated with the extent of deterioration of basal forebrain cholinergic neurons [20]. Notably, deep brain stimulation of the basal forebrain is being tested as a therapeutic option for dementia and can improve the cognitive symptoms of some Alzheimer’s and Parkinson’s dementia patients. Thus, progressive degeneration of central cholinergic neurons is thought to play a key role in neurodegenerative dementias and age-related cognitive decline, lending acute pathophysiological significance to basal forebrain research [21]. Meanwhile, recent studies have mentioned that the cholinergic system in aging can be affected by dietary supplementation, especially in the brain, which can be affected by natural substances and dietary components [22].

The purpose of this study is to examine whether roasted Sacha inchi has a harmful risk in HT22 cells. HT22 cells are originally immortalized from primary mouse hippocampal neuronal culture, possess functional cholinergic properties, and can be used as brain cholinergic neurons.

First, we determined extractant concentration for roasted Sacha inchi extract. Table 3 shows the result of HT22 cell viability in the hippocampus for validation of cognitive enhancement according to roasted Sacha inchi extract concentration (40 and 70% ethanol). It was found that HT22 cell viability treated with 10 μg/mL of roasted Sacha inchi extract from 70% ethanol (range = 98.60~102.39, mean = 100.83) is significantly (*p* < 0.001) higher than the control group (range = 97.77~98.89, mean = 98.45). In contrast, HT22 cell viability treated with 10 μg/mL of roasted Sacha inchi extract from 40% ethanol (range = 88.66~90.59, mean = 89.53) is significantly (*p* < 0.001) lower than the control group. A closer analysis of the treatment of roasted Sacha inchi extract from 70% ethanol shows significantly (*p* < 0.001) higher HT22 cell viability than the treatment of roasted Sacha inchi extract from 40% ethanol. This indicates that treatment with roasted Sacha inchi extract from 70% ethanol is suitable for the HT22 cell viability in the hippocampus for validation of cognitive enhancement.

Second, we determined the exposure time of roasted Sacha inchi extract from 70% ethanol. It was found that HT22 cell viability after 24 h (range = 98.60~102.39, mean = 100.83) is significantly (*p* < 0.001) higher than that after 48 h (range = 74.26~78.09, mean = 75.98). A close look at the exposure time 48 h shows significantly decreasing HT22 cell viability (*t*-value = 240.456). This indicates that treatment with roasted Sacha inchi extract from 70% ethanol for 24 h is suitable for the HT22 cell viability in the hippocampus for validation of cognitive enhancement.

#### 3.2.2. HT22 Cell Viability According to Extract Concentration of Roasted Sacha Inchi

HT22 hippocampal neuronal cells were exposed to roasted Sacha inchi at the concentration of 1, 5, 20, 40, 80, and 100 μg/mL for 24 h, and cytotoxicity was determined with the MTT assay. The cell viability obtained by the MTT assay for these concentrations is shown in Table 4 and Figure 2. The viability of cells exposed to the concentration of 1, 5, 10, 20, 40 and 80 μg/mL significantly increased to 105.61, 106.65, 105.87, 105.43, 103.69, and 104.21% of control (100.00%). In contrast, the viability of cells exposed to the concentration of 100 μg/mL was obtained only with no significant results, whereas no cytotoxicity was observed when the MTT assay was employed. As a result, roasted Sacha inchi has no cytotoxicity and has neuroprotective effects in a dose-dependent manner. These results suggest that roasted Sacha inchi may be a good source that exhibits neural protective activity.

Similarly, Nascimento et al. (2013) [23] found that the ethanolic fraction from Sacha inchi did not cause cytotoxicity in 3T3-L1 cells but did inhibit cell growth in two cancer cell lines (Hela and A549 cells). This indicates that the Sacha inchi extract may possess anticancer properties and is harmless to normal cells. Another study showed that raw Sacha inchi contained various levels of heat-labile phytotoxins, including alkaloids, saponins, and lectins, but effectively, roasting reduced these phytotoxins, and hence, thermal processing should be applied before the consumption of Sacha inchi [24].

However, there is a lack of study in cell viability of HT22 hippocampal neuronal cell, which has a cognition-enhancing property, and hence, a further study on this is expected to help find answers to the questions about the possibility of finding such cognition-enhancing effect. Further studies, including acute and chronic toxicity in animals, may also provide additional clarity for determining a safe intake level of Sacha inchi.

### 3.3. Development of Mousse Type Braised Sacha Inchi That Conform to Korean Standard for the Elderly

#### 3.3.1. Optimized Preparation Method of Braised Sacha Inchi

In this study, Sacha inchi braised in soy sauce and corn starch syrup which is called “Kong-jorim” in Korea, was converted to a mousse type product because it has a too hard texture to chew for the elderly.

Preference of sweetness, saltiness, and nuttiness scores of the experimental recipes ranged from 2.65 to 5.78, from 2.68 to 5.55, and from 2.75 to 5.89. To understand the effect of independent variables (soy sauce and corn starch syrup) on product preference scores, the regression model fitted to the experimental result of the preference of sweetness, saltiness, and nuttiness scores (Table 1) showed the model had an F-value of 204.16, 178.23, and 147.82. The *p*-value for lack of fit was <0.0001, <0.0001, and 0.0001.

Preference of overall taste scores from the elderly tasting panel of the experimental recipes ranged from 2.76 to 6.08. For understanding the effect of independent variables (soy sauce and corn starch syrup) on product overall taste preference scores, the regression model fitted to the experimental result of overall taste preference scores (Table 1) showed the model had a F-value of 24.80, which implies that the model is adequate. The *p*-value for lack of fit was 0.0041, which implies that the lack of fit was significant.

Described results can assert that the quality of the braised Sacha inchi is independent of a single factor. All independent variables are significant to defining the characteristics of the braised Sacha inchi, even if differently, depending on the specific response. Therefore, the next step involves the best combination of factors that can produce the expected characteristics of the final product. All comments arising from the response surface plots were taken into account in the optimization, considering that the optimal solution arises from a compromise among the different responses. In this phase, RSM software can be used to standardize the process under the desired operating conditions.

RSM gives the best recipe from different recipes after establishing not in doubt criteria or conditions for each factor for preparing good quality and acceptable food for the elderly. Kim and Joo (2015) [25] also successfully optimized food for the elderly, which is well-shaped and easy to chew and swallow, using RSM as applied to only consumer acceptability data using seven points hedonic scale from ten elderly’s food formulations. In contrast, this study employed four product responses to judge the food quality and optimization process. For the optimization process, the preferred criterion for each independent factor and response parameter was set from the numerical optimization menu displayed at the left-hand side of the central composite design (CCD). The criteria program will consider automatically but can be set manually. Preferably, conditions should be set manually with the objectives keeping in mind maximum nutrition, acceptability of consumers, and different taste preferences of the main population. In the present study, the goal of the experiment for the best recipe selection was to obtain braised Sacha inchi with soy sauce and corn starch syrup content in range, while the preference of sweetness, saltiness, nuttiness, and overall taste scores were kept to a maximum.

The program, which combined all the above-given criteria, generated ten solutions with the help of the “solutions” option on the left-hand side of the CCD design. However, No. 9 and 10 from Table 1 were selected with 45.00 g soy sauce and 25.50 g corn starch syrup showing the highest desirability and appeal. Thus, braised Sacha inchi recipe prepared with soy sauce and corn starch syrup, 39.00 g (2.60 tablespoon) soy sauce, and 23.29 g (1.37 tablespoon) corn starch syrup was selected for standardization with the highest desirability. The optimum solution No. 10 with maximum desirability from standard No. 10 in Table 1 indicated that the selected braised Sacha inchi recipe, when prepared with 39.00 g soy sauce and 23.29 g corn starch syrup, gave a predicted value of 3.92 sweetness preference, 3.94 saltiness preference, 3.91 nuttiness preference, and 3.90 overall taste preference score. This study optimized not only the overall quality of the braised Sacha inchi (an indicator of sensory acceptance) but also the chemical profile as well, which is essential to predict the nutritional quality of the braised Sacha inchi.

The present investigation can conclude that RSM can successfully apply for optimizing braised Sacha inchi recipes.The different levels of ingredients such as the amount of soy sauce (from 7.50 to 82.50 g) and corn starch syrup (from 8.5 to 42.5 g) employed for the preparation of braised Sacha inchi showed that these two variables markedly affected preference of sweetness, saltiness, nuttiness, and overall taste scores of the prepared braised Sacha inchi. The analysis using third-order polynomials that model *p*-values for all responses are less than 0.05 is significant at the 5% level, so it can be related to the ingredients’ levels. Results showed that sensory panelists generally responded with a high level of acceptance for braised Sacha inchi formulations containing 39.00 g soy sauce and 23.29 g corn starch syrup with the highest scores of 5.50~6.08 (out of 7). To select the best recipe/formulation, a recipe with 39.00 g soy sauce and 23.29 g corn starch syrup was chosen among ten recipes by using RSM, which can be used for numerical optimization options. Thus, the reason for using RSM in this study is that the braised sacha inchi recipe is optimized to provide the best formulation through various recipes, and it has the advantage of showing the effect of the ingredients(soy sauce and corn starch syrup contents) used on the food. These results can further help in developing newer and qualitative food for the elderly.

#### 3.3.2. Hardness of Mousse Type Braised Sacha Inchi

One of the principal quality parameters that determine the sensory features of food for the elderly is texture. It has complex physical features due to its structure and cohesive particles. Additionally, the Korea Ministry of food and rural affairs has quantified the chewable hardness standards of elderly food in three-level. Tenderization is necessary before Sacha inchi can be optimally utilized as a food resource for the elderly. However, there is minimal information in the literature about this aspect, even though aging populations present a growing consumer market within which Sacha inchi could play an important and necessary role. Accordingly, the development of a Sacha inchi mousse boiled down in soy sauce can make an antioxidant-dense product that helps elderly consumers increase their antioxidants. Thus, the idea of making mousse-type products was conceived to tenderize Sacha inchi for elderly consumers because of the lack of previous reports.

Table 5 presents the texture parameters of the mousse type of braised Sacha inchi. The values of the hardness of mousse type of braised Sacha inchi with gelatin from 3 to 17% had chewable hardness (under 20,000 N/m^2^) for elderly’s tongue, whereas the Sacha inchi mousse with gelatin from 19 to 27% had chewable hardness (22,000~50,000 N/m^2^) for elderly’s gum. In addition, the data show that the values of the hardness of mousse type of braised Sacha inchi with gelatin from 29 to 41% had chewable hardness (55,000~500,000 N/m^2^) for elderly with teeth. And the evaluated linear regression equation is ‘Y = 2536.3X + 18,946’ (where X is the concentration of gelatin powder and Y is the hardness of mousse type of braised Sacha inchi).

Predicting by the result produced, the values of the hardness of mousse type of braised Sacha inchi with gelatin 15.36% have chewable hardness with the elderly’s tongue and gelatin 27.18% has chewable hardness with elderly’s gum. Moreover, the values of the hardness of mousse type of braised Sacha inchi with gelatin 204.61% have chewable hardness with teeth.

These instrumental measurements provide complementary information on texture. While shear force is a good measure of initial bite tenderness, TPA gives more detailed information on the textural characteristics of the products. For elderly consumers, the most important parameter is probably hardness. The information reported in this study for TPA suggests that fine grinding and gelification is a promising procedure in nuts tenderization, thus encouraging the elderly to increase their Sacha inchi consumption. This technique can be considered as an effective tenderizer of Sacha inchi cooked in soy sauce as a practical application. It is not only for the elderly consumers but also for other users in the marketplace by having tender foods available using Sacha inchi.

#### 3.3.3. Sensory Evaluation of Optimized Mousse Type Braised Sacha Inchi by the Elderly

Sensory properties of the optimized mousse type of braised Sacha inchi were evaluated. The sensory evaluation was conducted with one hundred consumers, among them between 65 and 70 years of age, where the evaluated parameters were sweetness, saltiness, nuttiness, overall taste, through a seven-point hedonic scale for optimized mousse type of braised Sacha inchi. The results of the sensory analysis showed that the preference score of sweetness, saltiness, nuttiness, and the overall taste was obtained with 4.73, 4.82, 5.78, and 5.50, and the preference score of nuttiness had the highest score of sensory properties. The results of this study showed that the mousse type of food for the elderly could be prepared by supplementing roasted Sacha inchi with increased sensorial characteristics. These optimum mixture ratios from consumer acceptability tests could be used as reliable basic data for standardizing recipes of mousse type of braised Sacha inchi in the Korean food market for the elderly. Allied to this fact, using Sacha inchi in the development of elderly food is extremely important since it may contribute to their increased use and add value to the Sacha inchi.

## 4. Conclusions

The achieved results suggested that Sacha inchi can become good or moderate dietary sources of important nutrients such as antioxidants through roasting treatment. Therefore, its consumption should be increased for a healthy diet considering the antioxidant activity of seed origin. It can be concluded that roasting and consuming sacha inchi can save time and effectively maintain and increase its nutritional value. The following studies focusing on the optimal roasting technique, temperature, and duration will be required. The results of this study also indicated that roasted Sacha inchi uniquely promotes HT22 cell viability. The relatively hard texture of sacha inchi compared to other foods is a negative factor that its intake considering nutritional benefits. However, as research on the development of mousse-type aged food that is easy to chew and swallow, the factor has been decreased. These results present opportunities to produce antioxidants fortified Sacha inchi with a soft texture and satisfying technological characteristics. This study can be used as basic research data on the technological properties of Sacha inchi that have been subjected to the roasting process.

Results also suggest that Sacha inchi applied to mousse type of elderly food conforms to KS as a favorable option amongst texture revised diets in the elderly population. The development of a mousse type of Sacha inchi food can improve the texture and make it easier for the elderly to consume. Therefore, such advancement can help raise the quality of Sacha inchi products and secure consumers. It could be in the Sacha inchi industry’s best interest to make certain all of the population’s textural favors are catered.

This has been an exploratory study; follow-up studies are needed to confirm the detailed alterations of the Sacha inchi hardness and to adapt the technology to the food industry. Furthermore, it is necessary to define tenderness values that can identify and accommodate elderly consumers. Therefore, there is a need for additional research to introduce guidelines to promote its industrial utilization.

## Figures and Tables

**Figure 1 foods-10-02948-f001:**
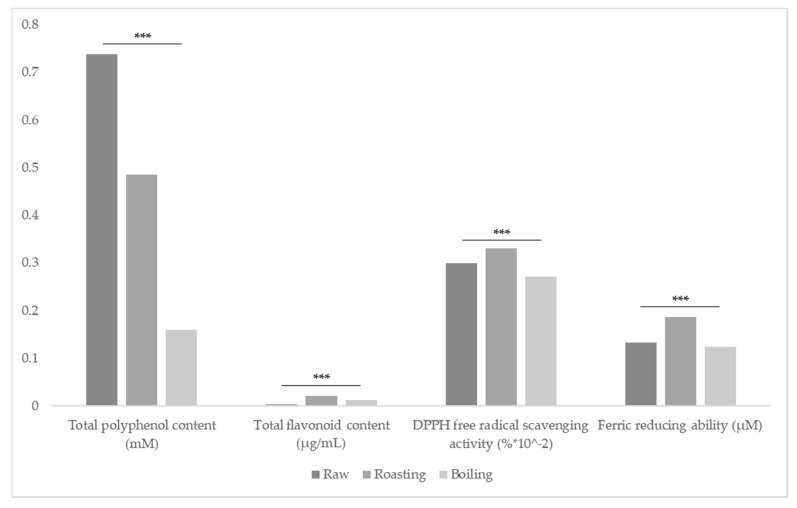
Analyzed antioxidant effects of raw, roasted, and boiled Sacha inchi. Data were subjected to one-way ANOVA followed by Tukey’s multiple comparisons using IBM SPSS statistics 23 (level of significance, *** *p* < 0.001).

**Figure 2 foods-10-02948-f002:**
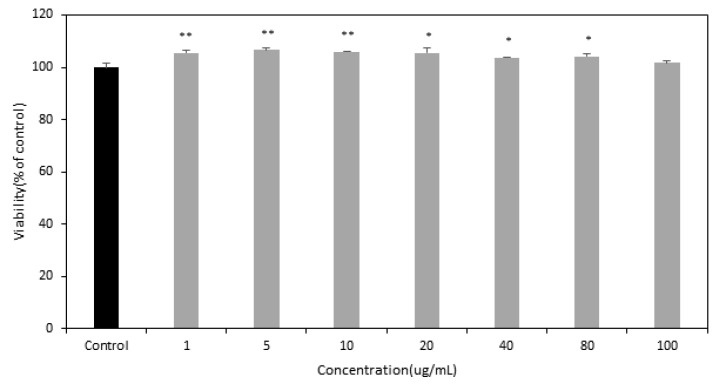
Effects of roasted Sacha inchi on HT22 cell viability. Cells were incubated with roasted Sacha inchi (1~100 μg/mL) for 24 h on confluent cells, and the viability was quantified by an MTT assay. Results of cell viability are expressed as percentage of untreated control cells. Results of three independent experiments were averaged and are presented as mean ± standard deviation. Data were subjected to one-way ANOVA followed by Tukey’s multiple comparisons using IBM SPSS statistics 23 (level of significance, * *p* < 0.05, ** *p* < 0.01).

**Table 1 foods-10-02948-t001:** Experimental design for response surface analysis and sensory properties of braised Sacha inchi.

Standard No.	Run No.	Soy Sauce (X_1_, g)	Corn Starch Syrup (X_2_, g)	Roasted Sacha Inchi (g)	Water (g)	Response			
						Sweetness (Y_1_)	Saltiness (Y_2_)	Nuttiness (Y_3_)	Overall Taste (Y_4_)
1	2	7.50	8.50	100.0	200.0	2.90 ± 0.14	2.79 ± 0.13	2.88 ± 0.14	2.98 ± 0.12
2	6	82.50	8.50			2.79 ± 0.13	2.68 ± 0.13	2.86 ± 0.14	2.97 ± 0.13
3	1	7.50	42.50			3.12 ± 0.14	3.34 ± 0.15	2.97 ± 0.13	2.95 ± 0.14
4	9	82.50	42.50			2.65 ± 0.13	2.97 ± 0.14	2.75 ± 0.13	2.76 ± 0.13
5	7	7.50	25.50			4.01 ± 0.20	4.32 ± 0.20	4.21 ± 0.21	3.91 ± 0.19
6	3	82.50	25.50			3.92 ± 0.18	4.11 ± 0.19	4.23 ± 0.20	3.72 ± 0.18
7	10	45.00	8.50			4.31 ± 0.21	4.21 ± 0.20	4.02 ± 0.19	4.13 ± 0.20
8	4	45.00	42.50			4.45 ± 0.22	4.30 ± 0.18	4.11 ± 0.20	4.20 ± 0.21
9	8	45.00	25.50			5.65 ± 0.27	5.50 ± 0.26	5.65 ± 0.28	5.79 ± 0.25
10	5	45.00	25.50			5.78 ± 0.26	5.55 ± 0.27	5.89 ± 0.29	6.08 ± 0.28
F-value (*p*-value)						204.17 *** (<0.0001)	178.23 *** (<0.0001)	147.82 *** (0.0001)	24.80 ** (0.0041)

Each value is an average of three replicates mean ± standard deviation. ** significant at *p* < 0.01, *** significant at *p* < 0.001.

**Table 2 foods-10-02948-t002:** Analyzed antioxidant effects of raw, roasted, and boiled Sacha inchi.

	Treatment			F-Value (*p*-Value)
	Raw	Roasting	Boiling	
Total polyphenol content (μM)	737.519 ± 60.47 ^a^	485.500 ± 10.89 ^b^	159.683 ± 7.56 ^c^	1051.063 *** (0.000)
Total flavonoid content (μg/mL)	0.004 ± 0.000 ^c^	0.021 ± 0.001 ^a^	0.012 ± 0.001 ^b^	377.444 *** (0.000)
DPPH free radical scavenging activity (%)	29.890 ± 1.09 ^b^	33.051 ± 2.08 ^a^	27.120 ± 0.88 ^c^	25.205 *** (0.000)
Ferric reducing ability (μM)	0.133 ± 0.002 ^b^	0.186 ± 0.008 ^a^	0.124 ± 0.003 ^b^	180.828 *** (0.000)

Each value is an average of three replicates mean ± standard deviation. *** significant at *p* < 0.001. ^a^, ^b^, ^c^ mean in a line by different superscripts are significantly different by Tukey’s multiple comparison method.

**Table 3 foods-10-02948-t003:** Analyzed HT22 cell viability after 24 h according to extractant concentration for roasted Sacha inchi.

Treatment	Percentage of Control
Control (DMEM)	98.45 ± 0.60 ^a^
Roasted Sacha inchi 40% ethanol extract	89.53 ± 0.98 ^b^
Roasted Sacha inchi 70% ethanol extract	100.83 ± 1.98 ^a^
F-value (*p*-value)	61.000 *** (0.000)

Each value is an average of three replicates mean ± standard deviation. *** significant at *p* < 0.001. ^a^, ^b^ mean in a row by different superscripts are significantly different by Tukey’s multiple comparison method.

**Table 4 foods-10-02948-t004:** HT22 cell viability in the hippocampus for validation of cognitive enhancement by roasted Sacha inchi.

Concentration (μg/mL)	Percentage of Control
Control (DMEM)	100.00 ± 1.54 ^c^
1	105.61 ± 0.84 ^a^
5	106.65 ± 0.90 ^a^
10	105.87 ± 0.10 ^a^
20	105.43 ± 2.12 ^a^
40	103.69 ± 0.32 ^ab^
80	104.21 ± 0.83 ^ab^
100	101.59 ± 0.97 ^bc^
F-value (*p*-value)	12.555 *** (0.000)

Each value is an average of three replicates mean ± standard deviation. *** significant at *p* < 0.001. ^a^, ^b^, ^c^ mean in a row by different superscripts are significantly different by Tukey’s multiple comparison method.

**Table 5 foods-10-02948-t005:** Hardness of mousse type braised Sacha inchi added with gelatin powder.

Concentration of Gelatin (%)	Hardness (N/m^2^)
3	844.67 ± 16.79
5	2013.78 ± 26.93
7	3235.49 ± 50.26
9	3883.55 ± 166.10
11	4656.08 ± 191.72
13	5757.47 ± 37.34
15	12,042.94 ± 292.71
17	15,764.71 ± 859.91
19	25,829.05 ± 428.53
21	31,152.57 ± 291.13
23	36,586.56 ± 329.20
25	43,441.47 ± 2105.53
27	49,551.54 ± 1519.17
29	54,008.06 ± 955.80
31	61,211.00 ± 1205.74
33	65,614.42 ± 1064.42
35	71,214.89 ± 342.37
37	78,184.04 ± 842.00
39	81,314.33 ± 1461.28
41	90,759.82 ± 1810.79

Each value is an average of three replicates mean ± standard deviation.

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
