# Peer review of "Feasibility of Elder-Friendly Food Applications of Sacha Inchi According to Cooking Method: Focusing on Analysis of Antioxidative Activity and Brain Neuron Cell Viability"

_foods, 2021, doi:10.3390/foods10122948_

Round 1

Reviewer 1 Report

The results are adequately presented, but the discussion of the paper generally lacks references to other research and an explanation of the results obtained. For example, it is necessary to highlight and explain, on the basis of other research and literature data, whether the heat treatment of raw materials has an influence on the bioavailability or absorption of the analyzed phytochemicals, as mentioned in relation to certain physiological conditions of the organism (possible chronic diseases, age, etc.) since the main topic of the paper was feasibility of age-appropriate food applications

The article by Authors Kim and Joo shows the potential in terms of research topics (antioxidant activity and brain neuron viability) and potential scientific interest, especially in terms of research on Plukenetia volubilis species, but the major shortcoming is the unclear and inadequately described plant material used, inadequate discussion, inadequately defined work objectives, a conclusion that needs to be completely restated.

Some of the specific comments are given below:

ABSTRACT
It is necessary to reformulate the abstract, it is too extensive and it is necessary to shorten it, take out one sentence of the introduction, the main results (the mentioned composition of the studied bioactive compounds) and the sentence of the conclusion. Interpret more clearly and highlight the objectives of the work.

INTRODUCTION
Lines 52-54 - Since the main topic of the paper is the antioxidants of the species Plukenetia volubilis, it is necessary to highlight more clearly what specific compounds it contains in order to give more examples.

MATERIALS AND METHODS 
Line 95 - More detailed information about the plant material used (e.g., at what state of maturity were the nuts used in the experiment, where was the plant material purchased, where did the plant material come from -origin, why was the plant material sorted by color and shape, what is the total amount of plant material used for the experiments).
Line 99-103 - There is a need to be more specific about the preparation methods used in the research rather than listing the reason for their use.Line 112 - Unnecessary chapter heading - please delete it.

CONCLUSIONS
The conclusion needs to be completely reworded, only the basic results should be referenced, the objective should again be avoided, after all the conclusion is a "response" to the objective of the paper, the conclusion does not cite any literature references. It is necessary to make the main results of the work shorter and clearer.

Reviewer 2 Report

Prezented  work was focused on the importance to perform a proper cooking for studying Sacha inchi (Plukenetia volubilis). The basic idea was to compare three typical states: raw, roasted, boiled. One of the objectives of this study was to determine the effect of cooking method on the antioxidant activity of Sacha inchi (Plukenetia volubilis). The results indicated that roasted Sacha inchi is distinguishable for its high content of antioxidants. The findings also indicated that Sacha inchi uniquely promotes HT22 cell viability.

The Authors carefully describe the aims and results of the study, however, there are some concerns should be addressed:

  • Abstract is too long and should be shortened.
  • No chromatographic analysis (qualitative and quantitative) of interested antioxidant compounds have been performed during experiment.
  • Authors should familiarize themselves with the proper format for references and make appropriate corrections.

Round 2

Reviewer 1 Report

Within the suggested suggestions, the authors have significantly improved the quality of the paper.